

# Atmospheric gas records from Taylor Glacier, Antarctica, reveal ancient ice with ages spanning the entire last glacial cycle

Daniel Baggenstos[1,*], Thomas K. Bauska[2], Jeffrey P. Severinghaus[1], James E. Lee[2], Hinrich Schaefer[3], Christo Buizert[2], Edward J. Brook[2], Sarah Shackleton[1], and Vasilii V. Petrenko[4]

[1]Scripps Institution of Oceanography (SIO), University of California, San Diego, La Jolla, CA 92093, USA.
[2]College of Earth, Ocean and Atmospheric Sciences, Oregon State University (OSU), Corvallis, OR, 97331, USA.
[3]National Institute of Water and Atmospheric Research Ltd (NIWA), PO Box 14901, Kilbirnie, 301 Evans Bay Parade, Wellington, New Zealand.
[4]Department of Earth and Environmental Sciences, University of Rochester, Rochester, NY 14627, USA.
[*]Current address: Climate and Environmental Physics, University of Bern, Switzerland.

*Correspondence to:* baggenstos@climate.unibe.ch

**Abstract.** Old ice for paleo-environmental studies, traditionally accessed through deep core drilling on domes and ridges on the large ice sheets, can also be retrieved at the surface from ice sheet margins and blue ice areas. The practically unlimited amount of ice available at these sites satisfies a need in the community for studies of trace components requiring large sample volumes. For margin sites to be useful as ancient ice archives, the ice stratigraphy needs to be understood and age models need to be established. We present measurements of trapped gases in ice from Taylor Glacier, Antarctica, to date the ice and assess the completeness of the stratigraphic section. Using $\delta^{18}O$ of $O_2$ and methane concentrations, we unambiguously identify ice from the last glacial cycle, covering every climate interval from the early Holocene to the penultimate interglacial. A high-resolution transect reveals the last deglaciation and the Last Glacial Maximum (LGM) in detail. We observe large-scale deformation in the form of folding, but individual stratigraphic layers do not appear to have undergone irregular thinning. Rather, it appears that the entire LGM-deglaciation sequence has been transported from the interior of the ice sheet to the surface of Taylor Glacier relatively undisturbed. We present an age model that builds the foundation for gas studies on Taylor Glacier. A comparison with the Taylor Dome ice core confirms that the section we studied on Taylor Glacier is better suited for paleo-climate reconstructions of the LGM due to higher accumulation rates.

## 1 Introduction

Ice cores from Greenland and Antarctica have provided high-resolution climate information over the past eight glacial cycles (EPICA Community Members, 2004). Many important climate parameters such as temperature, precipitation and greenhouse gas concentrations have been reconstructed using ice core measurements, providing a long-term perspective on modern climate change and a testing ground for climate models. However, deep ice cores suffer from severe limits on the volume of available ice from a given time horizon, precluding measurements that require large sample volumes (e.g. Petrenko et al., 2006, 2009). Blue ice areas, where ancient ice is brought to the surface by ice flow, and especially continental ice margins have been recognized





as valuable, inexpensive archives of paleo-climate information, which are not encumbered by sample size restrictions (Sinisalo and Moore, 2010; Petrenko, 2013).

In a steady state regime, the mass balance surplus on top of the Antarctic ice sheet is compensated for by ice flow towards its margins and eventually ice loss, through iceberg calving and melting at the base of ice shelves (Depoorter et al., 2013).

However, in certain places, ablation due to strong sublimation and low ice velocities combine to remove the young ice and expose old ice at the surface. The conditions needed for such outcropping of ancient ice are typically orographical rain shadows and/or flow stagnation because of obstacles such as nunataks or mountain ranges. A detailed review of blue ice areas and their specific meteorological and glaciological settings can be found in Bintanja (1999).

Historically, blue ice areas have been of interest mainly as meteorite traps (Whillans and Cassidy, 1983), but in the last

decade, some blue ice sites have been investigated for paleo-climatic information. In Antarctica, this includes sites at Scharffenbergbotnen (Sinisalo et al., 2007), Allan Hills (Spaulding et al., 2013), Mount Moulton (Custer, 2006; Korotkikh et al., 2011), Yamato Mountain (Moore et al., 2006), Taylor Glacier (Aciego et al., 2007), and Patriot Hills (Turney et al., 2013; Fogwill et al., 2017). Depending on the initial accumulation rate and the ice dynamics at a specific site, such records can span anywhere from a few thousand years at Yamato Mountain to 1 million years at Allan Hills (Higgins et al., 2015).

The main challenge in blue ice settings is age control. Traditional ice coring on domes or ridges is primarily a one-dimensional problem with ages increasing monotonically with depth. Similarly, at ice margins, one can expect the oldest ice at the very edge of the ice sheet and increasingly younger ages as one approaches the snow/ice transition. In principle, a monotonic undisturbed sequence of layers can be found on the margin just as it was deposited on top of the ice sheet (Reeh et al., 1991). However, deformation along the flow path from the ice sheet interior to the margin, typically close to the bed, can

lead to non-uniform thinning, tilting, folding, and faulting of the stratigraphic layers, complicating the dating and interpretation of the strata. Most previous work has relied on radiometrically dated ash layers (Dunbar et al., 2008) or stable water isotopes (Aciego et al., 2007; Spaulding et al., 2013) to establish ages for the outcropping ice. In recent years advances in measuring rare noble gas isotopes have led to new radiometric dating techniques applied to the trapped gas in the ice (Bender et al., 2008; Buizert et al., 2014; Yau et al., 2016). Common to all radiometeric dating techniques are relatively high uncertainties

which exclude these methods from high-resolution dating. Stable water isotopes, as used by Aciego et al. (2007) to determine a pre-Holocene age for most of the Taylor Glacier ablation zone, are a robust dating tool, able to identify rapid transitions and long term changes in the climate under which the ice was formed, but they do not provide unique age markers. Taking advantage of the trapped gas content in glacial ice allows for a more precise and accurate dating of the ice, due to the fact that many gases are globally well mixed and therefore must be the same in all trapped air sections of the same age (Bender et al.,

1994). Changes in the concentration and isotopic composition of globally well-mixed trace gases are continuously recorded in trapped gas bubbles as new ice forms. If a reference record for such gases from a well-dated deep ice core is available, it is possible to value-match an undated record to it and to uniquely identify the age of the ice using a suitable combination of gases (Blunier et al., 2007). In this study we use methane concentrations and molecular oxygen isotopic composition ($\delta^{18}O_{atm}$) and match them to the same tracers measured in the WAIS Divide ice core, a deep core from West Antarctica (WAIS Divide Project

Members, 2013, 2015).



Methane and oxygen are an ideal combination for this purpose because their average atmospheric lifetimes are very different; ~1,000 years for oxygen, but only ~10 years for methane. Methane concentrations respond rapidly to changes in methane sources and sinks, leading to large and fast variations, which provide precise age matching tie points (e.g. Rhodes et al., 2015). Oxygen isotopic content ($\delta^{18}O_{atm}$) varies more slowly but contributes unique information to the synchronization, especially

at times when methane has many similar sized fast variations or none at all (Bender et al., 1994). The combination of the two gases allows us to unambiguously date ice from most time intervals, making it a powerful dating tool. This method has been used previously to synchronize the time scales of deep ice cores (Malaize et al., 1994; Capron et al., 2010) and was also successfully used at the Pakitsoq ice margin in West Greenland to date a sequence of layers covering the last deglaciation (Petrenko et al., 2006; Schaefer et al., 2009).

The near-surface environment, which can alter the composition of the trapped gases through contamination with modern air, poses another challenge specific to ice margin sites. Most blue ice surfaces in Antarctica are riddled with a mosaic of thermal contraction cracks, formed by winter cooling and diurnal temperature changes in the summer, which provide a pathway for modern air to invade several meters deep into the ice (Popp, 2008). Gas analyses from near-surface (<1 m depth) ice from the Patriot Hills blue ice area were inconsistent, likely because of a combination of modern air contamination and microbial

production (Turney et al., 2013). Very old zero-ablation sites have been shown to have crack-induced contamination as deep as 33 m (Yau et al., 2015). Rapid ablation ameliorates this contamination problem to some extent, by removing the contaminated ice (Petrenko et al., 2009).

The ablation zone of Taylor Glacier (TG) is an ideal site for this study for the following reasons: (1) The glacier's flow field, sublimation rates and other glaciological parameters are well known (Robinson, 1984; Kavanaugh and Cuffey, 2009;

Kavanaugh et al., 2009a, b; Bliss et al., 2011); (2) a deep ice core was drilled near the deposition site for Taylor Glacier ice on Taylor Dome (TD) in 1993/1994, a mere 200 km upstream of Taylor Glacier, providing a point of reference and comparison (Steig et al., 2000; Morse et al., 2007); (3) it is a climatologically interesting area because of a controversial study by Steig et al. (1998) that argued that deglacial warming at Taylor Dome was in phase with Greenland, rather than Antarctic, climate; (4) it is within helicopter reach of McMurdo station with long periods of stable weather in summer, allowing for a logistically

simple field operation; and (5) previous work by Aciego et al. (2007) has shown that tens of kilometers of Pleistocene ice are exposed at the surface.

In this paper we present records of atmospheric composition from ancient ice outcropping in the ablation zone of Taylor Glacier. The aim is to date the ice and study the deformation it has acquired on its travel path, to assess archive integrity. At this site virtually unlimited amounts of old ice can be collected at the surface, allowing for the application of new proxies that have

hitherto been precluded by sample size restrictions (Buizert et al., 2014; Petrenko et al., 2016), and increasing the precision of established measurements that are hindered by small sample volumes. Two studies investigating the isotopic composition of $N_2O$ and $CO_2$ covering the deglaciation from Taylor Glacier are already published (Schilt et al., 2014; Bauska et al., 2016). Here, we provide the detailed age model for these studies and expand the temporal framework from the deglaciation to the entire last glacial cycle. Section 2 offers a description of the sampling strategy and the laboratory analyses. In Section 3 we show

that the trapped gas records are well preserved and contain a wealth of paleo-climate information. We describe two sampling



lines of late Pleistocene-age ice spanning the last glacial cycle. We discuss the stratigraphic layering and the observed folding. Finally, we present a high-resolution age model for 8 to 55 ka BP and highlight the differences of Taylor Glacier and the Taylor Dome ice core in the characteristics of their firn columns during the deglaciation and the Last Glacial Maximum (LGM).

## 2 Study area and methods

The Taylor Glacier ablation zone has been described in detail by Aciego et al. (2007) and Kavanaugh and Cuffey (2009): Briefly, ice flows from its deposition site on the northern flank of Taylor Dome through the Transantarctic Mountains into Taylor Valley (Fig. 1a). Ice velocities are on the order of 10 m yr$^{-1}$ in the center, and decrease towards the lateral margins where the glacier is frozen to the bedrock. Sublimation is 10–30 cm yr$^{-1}$ over most of the ablation zone, with higher rates at the base of the steeply sloping Windy Gully where winds speeds are highest, and with enhanced ablation close to the terminus due

to summer melting (Bliss et al., 2011). The ice surface is nearly horizontal and mostly free of crevasses. Thermal contraction cracks cover the entire surface of the glacier. Most of these cracks are confined to the surface, with a few reaching 4 m depth, and none were observed having propagated deeper than 7 m. There are numerous cryoconite holes with wind-blown sediment on the surface. The air bubbles in the ice are typically elongated, a product of their deformation history (Alley and Fitzpatrick, 1999).

Ice samples for gas analyses were collected during the 2009/10, 2010/11, 2011/12, and 2013/14 field seasons. We present data from two sampling lines, one that parallels the flow direction, and one that lies perpendicular to it (Fig. 1b). On the former, hereafter called the along-flow profile, 98 samples were collected over 20 km following the hypothetical center flow line, defined as the trace of maximum velocity based on measurements by Kavanaugh et al. (2009b). The sampling resolution increases from 1 sample per 500 m at the upstream end to 1 sample per 100 m close to the terminus because we expect the

strata to be increasingly more compressed in the downstream direction. On the $2^{nd}$ sampling line, hereafter called the across-flow transect, 300 samples were collected spanning 700 m in distance, in varying spatial resolution (from 1 m to 10 m), and approximately perpendicular to the along-flow profile. All samples originate from 4.0 m to 5.5 m depth and were carefully examined to be free of fractures that occasionally exist at this depth. The methane and $\delta^{18}O_{atm}$ data from the along-flow profile were presented in Buizert et al. (2014), but they are discussed here in much greater detail. In addition, three cores

were drilled to 12 m depth in different locations (Fig. 1b) to examine if the near-surface environment has an effect on the gas composition and sample integrity. These cores were sub-sampled in high depth resolution for a total of 65 samples. Again, we took great care to avoid fractures in these samples. However, in the shallowest ∼2 m the fractures are too pervasive to sample only solid ice, such that most shallow samples do incorporate some fractures.

Locations for the along-flow profile and deep core samples were recorded via GPS, while locations on the across-flow

transect were established with measuring tape and a limited number of GPS positions for reference because the across-flow sample spacing (1 m in some sections) is smaller than the accuracy of our GPS unit (5 m). Marker poles left in the across-flow transect allowed us to revisit the same sampling line for several years. All drilling was done with a PICO shallow coring drill assisted with a Sidewinder electric power head provided by Ice Drilling Design and Operations (IDDO, Koci and Kuivinen,



1984). After drilling and logging, the ice samples were stored for up to two weeks on site in an ice cave at approximately -10 °C in the 2009/10 season and in chest freezers at -24 °C in the following seasons, before being transferred to McMurdo Station.

For the purpose of establishing an age scale, we decided to tie the Taylor Glacier gas records to the WAIS Divide (WD) ice core. The WD core is described in detail elsewhere (WAIS Divide Project Members, 2013); for our purposes it is an ideal reference core because of its high-quality, high-resolution gas records (Marcott et al., 2014; Rhodes et al., 2015) and precise dating (Buizert et al., 2015). For ages older than 70 ka, which are not covered by the WD core, we are using the gas records from EDML as a reference (Capron et al. (2010) for $\delta^{18}O_{atm}$, Schilt et al. (2010) for methane).

The analytical method for measuring $\delta^{18}O_{atm}$ and $\delta^{15}N$ at Scripps Institution of Oceanography (SIO) has been described by Petrenko et al. (2006). The samples were measured on a Finnigan MAT Delta V dual inlet mass spectrometer. Results are presented with respect to modern air sampled at the end of the Scripps pier. The raw data were corrected for pressure imbalance and chemical slope (for $\delta^{18}O$, $\delta^{15}N$, and $\delta Ar/N_2$) using established analytical corrections. Following Bender et al. (1994) and Sowers and Bender (1995), we apply the gravitational correction to $\delta^{18}O$ using measured $\delta^{15}N$:

$$\delta^{18}O_{gravcorr} = \delta^{18}O_{measured} - 2 \times \delta^{15}N_{measured} \tag{1}$$

For all Taylor Glacier samples, a gas loss correction was applied as described by Severinghaus et al. (2009): Observed anomalies in $\delta O_2/N_2$ and $\delta Ar/N_2$ are used to correct $\delta^{18}O_{gravcorr}$ for inferred gas loss according to:

$$\delta^{18}O_{atm} = \delta^{18}O_{gravcorr} + a \times \left[\delta O_2/N_{2,gravcorr} + c\right] + b \times \delta Ar/N_{2,gravcorr} \tag{2}$$

a and b are empirical coefficients determined by a multivariate regression of pair differences of $\delta^{18}O_{gravcorr}$ with respect to pair differences of $\delta O_2/N_{2,gravcorr}$ and $\delta Ar/N_{2,gravcorr}$. c is also an empirical coefficient that offsets $\delta O_2/N_{2,gravcorr}$ such that the average value of $\delta^{18}O_{atm}$ in the last 1,000 years becomes zero. We do not have any samples from Taylor Glacier for the last millennium, and therefore cannot tune c in this way. To facilitate synchronization to the WD $\delta^{18}O_{atm}$ data, we optimize c such as to get identical values for both records at distinct features, e.g. extrema or plateaus. For the 400 Taylor Glacier samples measured in duplicates, a and b were determined to be close to Severinghaus et al. (2009)'s coefficients (0.0119 and -0.0155 for Taylor Glacier compared to 0.0136 and -0.0130 for Siple Dome), suggesting that the same gas loss mechanism is at work. This gas loss correction was not applied to the WD samples since the empirical gas loss relation was developed for bubbly ice and therefore is unlikely to apply to gas in a clathrate state as are all WD samples older than 10 ka. However, the WD samples exhibit much less gas loss in general due to very good temperature control during drilling, transport, and storage (Souney et al., 2014), such that the quality of the data is essentially the same as gas loss-corrected Taylor Glacier data.

For Taylor Glacier, a total of 725 samples from 352 separate locations or depths were analyzed mostly in duplicates. Ice samples from the same location but drilled in different years show excellent reproducibility. Of the 725 samples 12 were rejected for objective reasons such as incomplete transfer of the gas or other manual errors. An additional 7 samples were rejected due to poor agreement with replicates from the same location/depth. The remaining 706 samples have a pooled 1-sigma standard deviation of 0.005 ‰ for $\delta^{15}N$ and 0.011 ‰ for $\delta^{18}O_{atm}$, which is identical to the analytical precision achieved

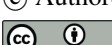



by Severinghaus et al. (2009). For the WD core, 169 samples from 147 depths were analyzed on the same instrument. The 21 duplicate samples have a pooled standard deviation of 0.003 ‰ for $\delta^{15}N$ and 0.010 ‰ for $\delta^{18}O_{atm}$.

Methane concentrations were measured in the field and in the lab at Oregon State University (OSU). The field system is a manual version of automated systems used in the OSU laboratory (Mitchell et al., 2011, 2013). Methane is measured
using a melt-refreeze technique, using glass vacuum flasks with Conflat flanges on a 12-port vacuum manifold pumped with a scroll pump. Concentrations are measured with respect to calibrated air standards, tied to standards maintained at the OSU laboratory that are calibrated by the NOAA Earth Systems Research Laboratory Global Monitoring Division, on the WMO X2004 concentration scale (Dlugokencky et al., 2005). As described in more detail in Mitchell et al. (2011, 2013) a gas chromatograph (Shimadzu GC-14 in the case of the field system) with flame ionization detector is used to measure methane,
and the concentrations are quantified by measuring the methane peak area and total pressure in the GC gas-sampling loop. The initial field measurements were corroborated by lab measurements, using the analytical method described by Mitchell et al. (2011, 2013). Corrections for solubility, blank size, and gravitational enrichment are applied. The precision of the field measurement of methane is estimated at 9 ppb based on a small number of replicates (n = 9); the laboratory measurements are precise at the 7 ppb level also based on replicates (n = 66). A comparison between field and lab analyses is described in
Appendix A.

## 3  Results and discussion

### 3.1  Reliability of the gas records

Figure 2 shows three high-resolution depth profiles of $\delta^{18}O_{atm}$ and methane concentration at different locations on the glacier. Overall, the methane profiles show more variability than the $\delta^{18}O_{atm}$ profiles. This is not surprising given the higher short term
variability of the underlying methane history, and the fact that methane as a trace gas is easier to contaminate than molecular oxygen as a major component of the air mixture. From 0 m to 4 m depth, clear and significant anomalies are apparent in the methane profiles. The $\delta^{18}O_{atm}$ profiles show such anomalies only in core D3. Since we observe a different contamination pattern in each core (methane and $\delta^{18}O_{atm}$ decreasing, methane decreasing while $\delta^{18}O_{atm}$ is stable, methane increasing while $\delta^{18}O_{atm}$ is stable) it is difficult to determine the exact processes responsible, but it is likely due to a combination of entrainment
of modern air through cracks and biological activity in cryoconite holes and in cracks (Stibal et al., 2012). Below 4 m there is no sign of alteration of any kind (the drop in methane values from 580 ppb to 470 ppb in core D3 is a paleo-atmospheric signal, as explained below, and not an artifact). This means that samples from deeper than 4 m are representative of the atmosphere when the air was trapped in the ice.

We can therefore use these data to assign ages to the different locations by comparing the gas composition to other ice cores.
We restrict the age range to the last glacial cycle, a well-justified assumption given that these sites are 10+ km upstream from a site that has been dated to 120 ka BP (before present = before 1950 AD) and close to two other sites that were dated to 12 ka BP and 50 ka BP by Buizert et al. (2014) using Kr-81 radioisotope dating. The gas data then suggest unique ages for cores D2 and D3, and two possible ages for D1 (Fig. 3a,d): For core D2, the combination of methane just under 400 ppb and



$\delta^{18}O_{atm}$ of 0.92 ‰ is only possible in the LGM, at ~20 ka BP. For D1, characterized by 450 ppb methane concentration and $\delta^{18}O_{atm}$ of 0.4 ‰, either an age of 35 to 45 ka BP or around 90 ka BP would satisfy the constraints provided by the two gases. In this case it is not possible to uniquely identify the age based on a single location in the transect. However, additional data from adjacent samples would likely preclude one of the two options and the proximity to D2 favors the younger age estimate.

For D3, a $\delta^{18}O_{atm}$ value of 1.1 ‰ is sufficient to uniquely identify the age as 15 ka BP. This is exactly the time of the Oldest Dryas–Bølling transition with a large increase in methane, which explains the large shift in the D3 methane profile at 5 m depth. The fact that methane decreases from 580 ppb to 470 ppb with depth therefore must mean that at this site the ice gets older with increasing distance from the surface, and is thus in the order in which it was deposited (i.e., not upside down). For D1 and D2, the constant methane and $\delta^{18}O_{atm}$ values with depth suggests that at these locations the ice either has constant age

with depth (ice layers dip close to vertical) or age variations are small enough to remain within periods where the measured parameters were stable.

A small offset between field and laboratory results is discussed in Appendix A; it does not indicate alteration of the paleo-atmospheric signals.

## 3.2   Along-flow profile covers entire glacial cycle

The along-flow profile covers 25 km of the ablation zone of Taylor Glacier, closely following the center flow line. As expected from first principles, the ice gets older towards the terminus (Fig. 3b): From 28 km to 24 km, methane concentrations of 650–750 ppb are a clear indicator of Holocene ages for these samples. In addition, increasing $\delta^{18}O_{atm}$ in the downglacier direction shows that the ice becomes older in that direction. From 23 km to 19 km, the gas records continue to indicate increasing age of the ice, capturing the deglacial sequence: Younger Dryas (YD, 460 ppb methane, 0.6 ‰ $\delta^{18}O_{atm}$), Bølling/Allerød (B/A,

650 ppb methane, 0.6–1.1 ‰ $\delta^{18}O_{atm}$), and Oldest Dryas (OD, 350–450 ppb methane, 1.1–1.0 ‰ $\delta^{18}O_{atm}$). At 19.3 km, a methane concentration value of <400 ppb is indicating an LGM age. The following section, 19 km to 13 km, is characterized by $\delta^{18}O_{atm}$ of 0.4–0.2 ‰ and variable methane concentrations, the combination of which is indicative of MIS 3, which spans 29 to 57 ka BP. Continuing further downstream, it is no longer possible to align the Taylor Glacier along-flow record with a deep ice core record by varying the Taylor Glacier age-distance relationship because $\delta^{18}O_{atm}$ decreases to 0.0 ‰ (possibly

MIS 5a) instead of increasing to 0.7 ‰ (MIS 4) as expected from the atmospheric history. Thus MIS 4 appears to be missing entirely. Still, $\delta^{18}O_{atm}$ alternating between 0.5 ‰ and -0.1 ‰, as seen from 13 km to 7 km, is consistent with MIS 5a–5d. Around 10 km there are two data points with a $\delta^{18}O_{atm}$ value of 0.7 ‰ which could indicate MIS 4 (and thus an age reversal), or MIS 5d. Finally, the very low $\delta^{18}O_{atm}$ values of -0.4 ‰ close to the glacier terminus are a clear indication of interglacial origin, in this case most likely Eemian (~125 ka BP). The Eemian age of this part of the glacier has already been independently

confirmed by Buizert et al. (2014) using Kr-81 dating.

This transect confirms that there are large amounts of Pleistocene-age ice outcropping on Taylor Glacier, in agreement with Aciego et al. (2007). The Oldest Dryas and LGM layers are significantly thinner than one would expect looking at the Siple Dome or WD records. However, it is consistent with the Taylor Dome record, which also has an anomalously thin LGM layer because of low accumulation rates (Steig et al., 2000; Morse et al., 2007). Hyper-arid conditions in the Taylor Dome region





were probably the result of a change in the trajectory of moisture bearing storms and increased distance to open water, because of the advance of the Ross ice shelf far into the Ross sea (Morse et al., 1998; Aarons et al., 2016).

A first order age model for the along-flow profile was constructed by identifying maxima, minima, and transitions in the $\delta^{18}O_{atm}$ and methane records that unambiguously belong to a certain time period, e.g. the $\delta^{18}O_{atm}$ minima of MIS 5a, 5c,
5e and the methane transitions at the beginning and end of the Younger Dryas. Even though the samples from the Aciego et al. (2007) study are not from the exact same locations, it is still instructive to compare their age/distance relation with the one from our along-flow profile (Fig. 4): For kilometer 28 to 20, the two age models are in good agreement with each other. Downstream of kilometer 20, our ages are consistently older than the Aciego et al. (2007) ages and the discrepancy increases towards the terminus. The most plausible explanation for this age discrepancy between the slightly offset sampling lines is
substantial across-flow age variability (Fig. 1b). The fact that the stratigraphy gets more and more compressed towards the terminus, as expected from a simple ice flow model but also evident in the age/distance relationship, can then explain why the age discrepancy increases moving downglacier.

### 3.3 Across-flow transect reveals deglaciation in high resolution

The same deglacial sequence identified in the along-flow profile is also found in the across-flow transect, albeit with a com-
plicating twist. The transect was purposely run across a large z-fold structure, visible in aerial images, to examine whether the surface expression of the ice (color, resistance to sublimation) is indicative of its stratigraphy. Crossing the limbs of a z-fold is predicted from structural geology theory to lead to a tripling of stratigraphic layers. Indeed, instead of one $\delta^{18}O_{atm}$ maximum, at the Oldest Dryas–Bølling transition as in deep ice cores, we observe three, along with an additional Bølling/Allerød interval, characterized by elevated methane values (Fig. 3c). The folding and age scale determination are discussed in detail below in
Sections 3.4 and 3.5. The youngest ice in this transect is of early Holocene age, while the oldest ice is approximately 60,000 years old based on $\delta^{18}O_{atm}$ of 0.2 ‰. Further sampling to the Southeast, extending the transect from -300 m to -500 m, yields largely unchanging $\delta^{18}O_{atm}$ (not shown). This suggests a change of geometry, with the stratigraphy changing from close to vertical to essentially horizontal; the ice outcropping at the surface is then of equal age. This hypothesis is corroborated by satellite imagery of the area which appears devoid of stratigraphic features seen in other locations on the glacier, and a deep
vertical core that found horizontal upright stratigraphy using gases. Finally, the $N_2O$ and $CO_2$ records from Schilt et al. (2014) and Bauska et al. (2016), respectively, confirm that the section spanning -124 m to +120 m on our sampling line (Fig. 3c) represents the last deglaciation, in excellent agreement with our $\delta^{18}O_{atm}$ and methane data. They also highlight the potential for high-quality, high-resolution gas studies on Taylor Glacier.

The whole sequence from the early Holocene well into MIS 3 is compressed into a few hundred meters on the transect,
whereas it takes almost 15 km in the along-flow direction to cover that same time span. The most logical explanation for this observation is that the strike of the outcropping layers, or isochrones, is parallel to the flow direction. Consequently, the largest change of age with distance is perpendicular to the glacier flow. Aciego et al. (2007) already noted that gradients of age in the cross-flow direction are likely to be much greater than in the along-flow direction. This interpretation is strongly supported by satellite imagery, airborne imagery, and observations on the ground, all of which show individual layers of different shades of





blue colors that run parallel to the flow direction (Fig. 5a). The color of the ice varies from almost white-blue to reddish-blue due to different amounts of dust in the ice, representing different climate states and thus chrono-stratigraphic layers.

This age distribution is a somewhat surprising result, since the intuitive expectation is that the age gradient is largest in the along-flow direction. Without knowing the full 3-D structure of the stratigraphy of the glacier it is difficult to reconstruct the ice dynamics and deformation history that led to the observed age pattern. However, we can hypothesize about the processes that could arrange the isochrones parallel to the flow direction. One possibility is that large velocity gradients in the flow field cause longitudinal shearing, which will eventually orient all in-glacial structures parallel to the flow direction, regardless of their original orientation. The Taylor Glacier ablation zone is narrow enough that the surface velocity is dependent on the distance to the margin for most of the area. Only in a narrow central band is true plug flow behavior observed (Kavanaugh and Cuffey, 2009). However, even close to the equilibrium line, 60 km from the terminus, where the ice from the polar plateau enters Taylor Valley, one can find such along-flow oriented features. At this point the glacier is much wider, which reduces the influence of lateral shearing, but it is still possible that large velocity gradients exist because of the bottom topography. Another explanation for the unusual pattern of stratigraphy and flow holds that layers deposited horizontally on the plateau are experiencing latitudinal compression upon entering a glacial trough or valley. This forces the layers to rotate by 90 degrees, forming a large syncline, which would produce outcropping stratigraphy that parallels the flow direction. In the end, resolving the entire deformation history from the complex interplay of bottom topography, accumulation history, surface flow and flow at depth is beyond the scope of this study. It would be interesting to test whether a state-of-the-art 3-D full-Stokes numerical glacier model can reproduce the outcropping age pattern. Finally, the large age gradient in the across-flow direction explains how Aciego et al. (2007)'s and our along-flow sampling line can have dramatically different ages despite their relative proximity.

## 3.4 Large scale folding

The most obvious stratigraphic feature is the layer of ice deposited in the LGM, with a characteristic reddish color due to high dust concentrations in the ice matrix (reddish algae dependent on the dust may create the characteristic color, Lutz et al., 2016). In the across-flow transect, this "dusty band" is encountered in two locations, from -110 m to -160 m, and around 0 m (Fig. 5a). The gas signature in both locations is the same, confirming that this is a single isochronous layer, which must be connected outside of the transect line or in the subsurface. Indeed, a short distance downstream the two limbs connect in a large folded structure, indicating a plunging fold with a nearly horizontal fold axis oriented in the along-flow direction. The fold is easily visible on airborne (Fig. 5a) and satellite imagery (Google Maps, 2015). It is a z-fold, composed of a synform/antiform pair, with the axial planes approximately vertical and parallel to the flow direction. The LGM layer in the north-western limb is visible at the surface for the first time at the across-flow transect, and continues to emerge in the downglacier direction at a shallow angle. The syncline (south-eastern part of the z-fold, with young layers in the center of the fold) also emerges at the same shallow angle, and disappears from the surface a few kilometers downstream, exposing the older layers underneath it.

We estimated the dipping angle of the layers at a few locations in the transect by drilling vertical profiles and comparing the data to the surface sequence. For the "young" side (north-western part of the anticline, see Fig. 5b), the layer dip is 70° to 80°



(Petrenko et al., 2017), but decreases as expected with increasing proximity to the fold axis. On the "old" side (south-eastern part of the syncline), the layer dip distant from the fold axis is approximately 90°, i.e. vertical. The resulting fold geometry is pictured in Figure 5b. It shows the multiplication of a single stratum in undisturbed stratigraphy into three limbs in the folded part as seen in our gas data: The upright limb of the syncline, the overturned limb that is part of the syncline and anticline, and

the upright limb of the anticline.

Folding and deformation is very much expected in a fluid medium like glacial ice. Folds can easily form in a variety of settings, as long as there are velocity gradients which induce shearing (Jacobson, 2001) or layers of differing rigidity. Indeed one can find deformation patterns in many parts of Taylor Glacier, ranging in scale from centimeters to hundreds of meters, typically in the form of z-folds. It is not clear where these folds form, with previous research showing that some folds originate

upstream of the region of streaming flow in a West Antarctic ice stream (Jacobel et al., 1993). In structural geology, z-folds are usually viewed as minor folds in the limbs of larger synform or antiform structures (Bell, 1981; Hudleston and Treagus, 2010). In this case, this interpretation would imply a large syncline with the layers folded up symmetrically to fit the "U-shaped" valley.

### 3.5 An age model based on gas synchronization

The across-flow transect is the preferred sampling line for the deglaciation, LGM, and MIS 3 time periods because it is oriented perpendicular to the stratigraphic order. The along-flow profile runs almost parallel to the strata, which can easily lead to spurious variations if the sample path deviates from the true flow direction even slightly. We developed a high-resolution age model for the across-flow transect using our gas data and a dynamic programming algorithm developed by Lisiecki and Lisiecki (2002) to correlate the paleo-climate records in an objective and reproducible fashion. Sharp methane transitions and

inflections in the $CO_2$ record (Bauska et al., 2016) provide fixed tie points (Table 1) and the computer algorithm value matches the $\delta^{18}O_{atm}$ data in between tie points. The algorithm minimizes the difference between two data series while observing fixed tie points and discouraging large accumulation rate changes, i.e. the resulting distance/age curve is as smooth as possible. The model cannot optimize both the fit to the data and the smoothness of the distance/age curve, thus a parameter is subjectively chosen that produces an acceptable tradeoff (called the speed change penalty in the Lisiecki and Lisiecki (2002) model). To

assess the uncertainty in our age estimates, we apply a Monte Carlo randomization scheme by running the model 5,000 times while resampling the $\delta^{18}O_{atm}$ input data subject to its analytical error. Uncertainty in the exact location of the methane and $CO_2$ tie points in our transect is also taken into account by resampling the location of the tie points in a range determined by the sample spacing (Table 1). The resulting uncertainty is a lower bound of the total uncertainty, since the tradeoff between optimizing the model fit and the smoothness of the distance/age relationship is not randomized. Still, changing the tradeoff

parameter will only affect the fine scale details of the model, and should not contribute significantly to the overall uncertainty. Because of the fold in the stratigraphy, we create two separate age models, one each for the "old" (-80 m to -300 m) and "young" (30 m to 260 m) side of the fold. The two age models overlap by approximately 2,000 years.

Figure 6c/e shows the resulting distance/age relationship as well as the age uncertainties. The distance/age curve looks surprisingly similar to depth/age curves from deep ice cores with older layers being more compressed than younger ones,



which is equivalent to a flattening in the depth/age curve. Non-uniform thinning, which has been described as a consequence of deformation at the West Greenland ice margin (Petrenko et al., 2006), does not seem to have significantly affected this sequence of layers. The uncertainties in our age model are directly related to the underlying gas histories. The uncertainty is small whenever there are abrupt methane transitions or $\delta^{18}O_{atm}$ is changing rapidly, providing good age control. Therefore,

in the "young" section, the uncertainty is generally less than 500 years, thanks to a wealth of tie points and large changes in $\delta^{18}O_{atm}$. In the "old" section, age uncertainties are substantially larger because of fewer methane tie points and long periods of unchanging $\delta^{18}O_{atm}$. For ice older than 47 ka BP (early MIS 3) the age model is poorly constrained because of low sampling resolution, which is reflected in elevated age uncertainties.

### 3.6   Taylor Glacier vs. Taylor Dome

After placing the Taylor Glacier across-flow transect on a time scale, we can compare its environmental records to deep ice cores, with Taylor Dome being the most logical and interesting choice due to its proximity to the Taylor Glacier deposition site. $\delta^{15}N$ of $N_2$ of air trapped in ice is enriched compared to atmospheric values due to gravity-driven diffusive separation in the firn column (Craig et al., 1988; Sowers et al., 1989). The enrichment is nearly linearly related to the thickness of the diffusive column, and is therefore used as a proxy for firn thickness, although it is also affected by firn temperature gradients

(Severinghaus et al., 1998) and convective air mixing in the shallowest firn (Kawamura et al., 2006). Comparing $\delta^{15}N$ from Taylor Dome gas records and Taylor Glacier gas records highlights interesting differences (Fig. 6d): During the LGM ($\sim$20 ka BP), gravitational enrichment as seen in $\delta^{15}N$ almost ceases ($\delta^{15}N \approx 0$) at Taylor Dome, suggesting a very thin firn and possibly deep air convection, both of which are only possible with an extremely low accumulation rate (Severinghaus et al., 2010). Taylor Glacier $\delta^{15}N$ is also low but never falls below 0.07 ‰, indicating a thicker, more typical firn column during this

climate period, most likely due to a higher accumulation rate. In addition, the layer thickness of the dusty band, representing the LGM, at Taylor Glacier is $\sim$50 m thick, whereas it is only $\sim$10 m thick in the Taylor Dome ice core, using measured dissolved calcium concentration as a proxy for dust (Steig et al., 2000). Since the layer dip is nearly vertical on the old side of the fold, the distance measured on the surface is approximately equal to the true layer thickness. It is likely that the LGM layer in Taylor Dome has been subjected to more thinning than the layers that are exposed on Taylor Glacier, but thinning alone cannot explain

the five-fold difference in observed thicknesses. Instead, the differing layer thicknesses suggest that there was a steep gradient in accumulation rate across Taylor Dome. This conclusion is in agreement with radar profiles (Morse et al., 1998) that show that the LGM layer is significantly thicker on the northern flank of Taylor Dome (the deposition area of Taylor Glacier ice) than where the deep ice core was drilled, because of predominantly northerly storm trajectories. This view is further strengthened by more recent work from Morse et al. (2007) that confirms that the accumulation rate during the LGM at the Taylor Dome

ice core site was even lower than previously estimated. All of the above makes Taylor Glacier more suitable for paleo-climate reconstructions of the LGM than the original deep ice core.



## 4  Conclusions

Taylor Glacier is the first ice margin site in Antarctica where paleo-atmospheric gas records from the last glacial cycle have been explored in detail. The gases are well preserved below 4 m depth and contain unaltered climate information. The combination of $\delta^{18}O_{atm}$ and methane concentration allows for unambiguous identification of different climatic intervals in the ice through

comparison to deep ice core records. Using this approach, a complete sequence of ice ranging from 8 to 55 ka BP has been identified. While broad folding of ice layers has been observed in some of the sampled ice sections, this has not been an obstacle to obtaining good quality, well-dated gas records.

Our age mapping of outcropping ice builds the foundation for future work on Taylor Glacier. The development of this easily accessible archive of ancient air (and ice) may enable studies that until now have been limited by availability of large ice

volumes, complementing traditional deep ice coring efforts. Furthermore, a comparison to Taylor Dome shows that Taylor Glacier is better suited for paleo-climate reconstructions of the LGM than the Taylor Dome ice core because the ice was deposited at higher accumulation rates. Finally, the age of the ice at the Taylor Glacier terminus is of interest to biologists studying the microbial community in Lake Bonney, a perennially frozen lake formed by Taylor Glacier outflow (Fountain et al., 1999). We have confirmed that the very oldest ice outcropping at the terminus is substantially older (125+ ka BP) than

previously estimated.

Understanding the stratigraphy in the lower part of the ablation zone is still a work in progress. We are expecting more deformation in this very old ice because the ice has travelled close to the bedrock. Unfortunately, remote sensing analysis of large scale structures in the stratigraphy is hampered by a network of melt channels that obscure the stratigraphic signature of the ice on this lowermost part of the glacier. The three dimensional stratigraphic pattern identified in this study presents

potentially useful information to glaciologists studying ice dynamics and deformation processes. It also presents a challenge to explain the mechanics that led to the observed stratigraphy. Future work on the across-flow transect could include increasing the sampling resolution to 1 m for the oldest part, and additional depth profiles that would constrain the layer dip in more places.

## 5  Data availability

The data will be submitted to the Antarctic Glaciological Data Center (AGDC) at the National Snow and Ice Data Center (NSIDC).

## Appendix A:  Field based versus laboratory methane measurements

Methane measurements both in the field and in the lab of identical samples reveal a slight offset with the field data being elevated compared to the lab data. The offset was likely caused by a higher blank in the field system related to the difficulty

to keep the glass vessels as clean as possible. Using replicate measurements the offset was quantified as 11 ppb (+/-16 ppb 1-sigma s.d., n = 55) for the 2010/11 season, 22 ppb (+/- 14 ppb 1-sigma s.d., n = 2) for the 2011/12 season, and 17 ppb (+/-



18 ppb 1-sigma s.d., n= 22) for the 2013/14 field season. Unfortunately, the offset is poorly defined and likely underestimated by only two replicates during the 2011/12 season when most of the samples older than 20 ka were drilled and analyzed. The elevated methane values (compared to WD) from 20 ka to 35 ka BP (Fig. 6a) are thus likely a contamination artifact of the field based extraction system. There is no evidence that the gas content is altered in ice older than 20 ka BP, as demonstrated

5 by the excellent agreement of lab based methane measurements with the WD reference (Fig. A1).

*Competing interests.* The authors declare that they have no conflict of interest.

*Acknowledgements.* We thank Kurt Cuffey for helpful discussions on glaciology and the logistics of living on Taylor Glacier. The US Antarctic Program provided outstanding logistical support. IDDO (Ice Drilling Design and Operations) provided the drilling systems. The Polar Geospatial Center (PGC) provided satellite imagery. We thank Kate Koons, Chandra Llewellyn, Paul Rose, Bija Sass and Martha Story

10 for keeping everybody safe, fed, and happy. Michael Dyonisius, Xavier Fain, Ben Hmiel, Tanner Kuhl, Robb Kulin, Logan Mitchell, Avery Palardy and Adrian Schilt all helped with sampling in the field. We thank Ross Beaudette for laboratory assistance. This work is supported through NSF Polar Awards 0839031, 1246148, 1245821 and 1245659. Further support came from the Marsden Fund Council from New Zealand Government funding, administered by the Royal Society of New Zealand and NIWA under Climate and Atmosphere Research Programme CAAC1504 (HS).



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



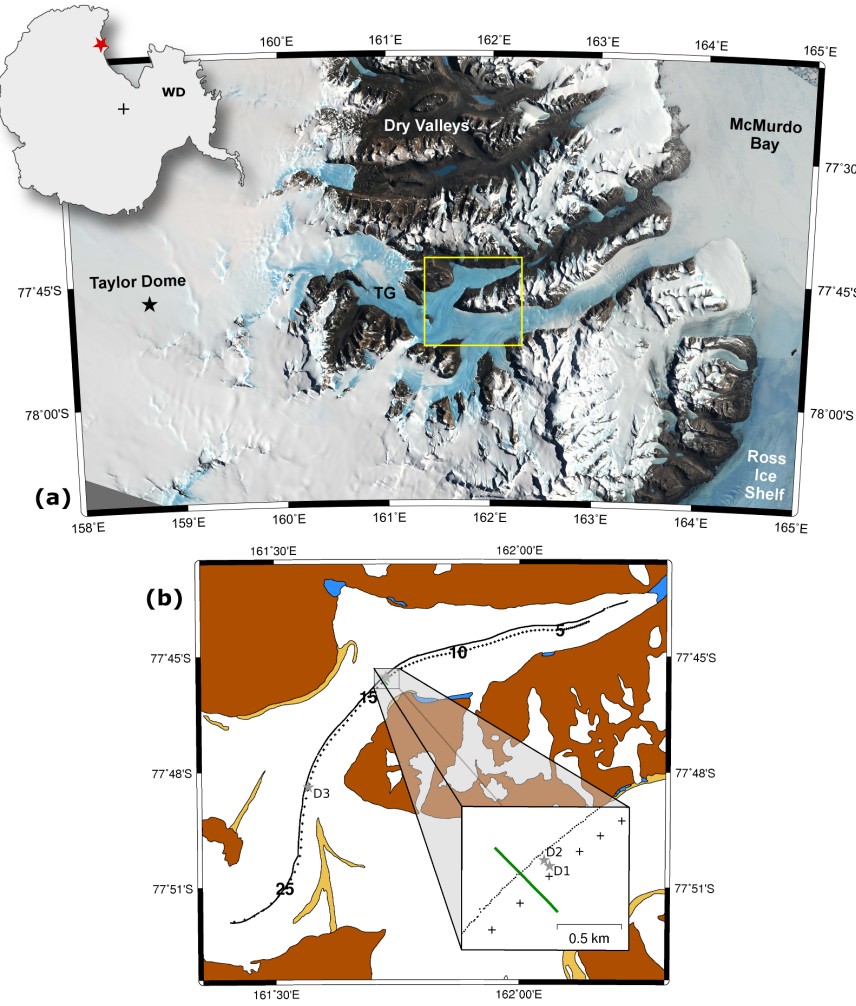

**Figure 1. (a)** Map of the McMurdo Dry Valleys, with Taylor Glacier (TG) in the center. The Dry Valleys are a series of valleys in the Transantarctic Mountains within Victoria Land (red star). Ice flowing into Taylor Valley originates on the northern flank of Taylor Dome. Satellite imagery courtesy of the U.S. Geological Survey (Bindschadler et al., 2008). **(b)** More detailed map of the Taylor Glacier ablation zone including the sampling lines. Aciego et al. (2007)'s samples (·) and our along-flow profile samples (+) are shown. Each symbol represents one sample, but the high sampling resolution causes the Aciego et al. (2007) samples to blend into a solid line, whereas our samples appear as dots. Numbers on the profile denote approximate distance from the glacier terminus in km. The inset shows an enlarged view of the across-flow transect. The solid green line marks the across-flow transect. Gray stars indicate the positions of the three deep cores D1–D3. Ice flow is from the SW to the NE (lower left to upper right). Taylor Glacier receives a small amount of ice from Ferrar Glacier (lower right), with a distinct suture (not shown) separating the two ice masses.





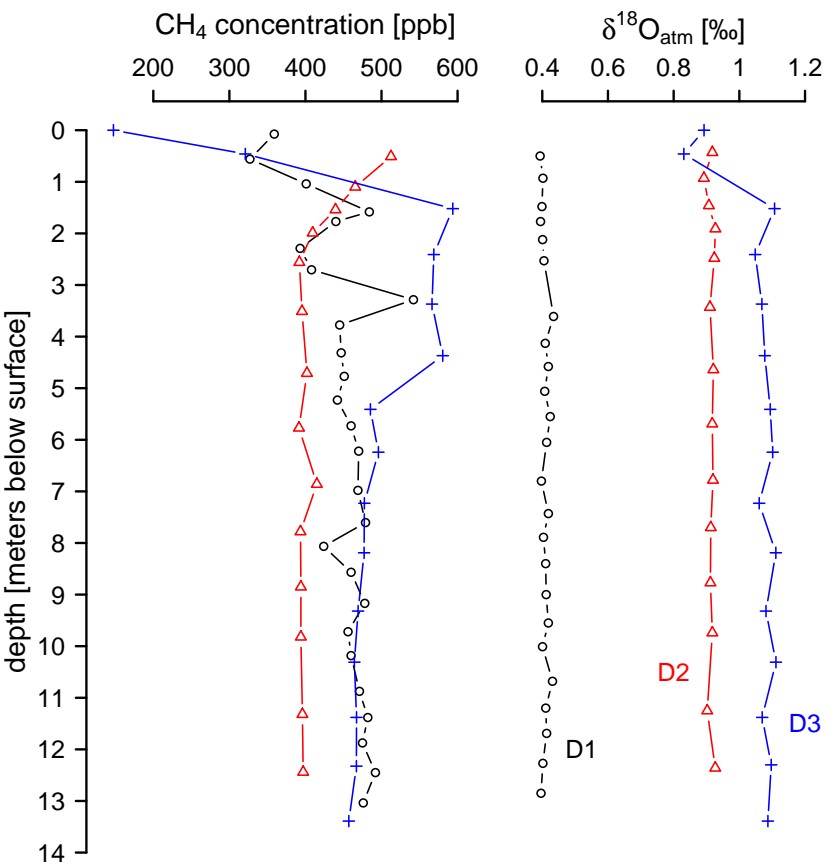

**Figure 2.** Depth profiles of methane and $\delta^{18}O_{atm}$ at the sampling sites D1 (○), D2 (△), and D3 (+). The analytical uncertainty based on replicate analyses is smaller than the size of the symbols in the plot for both methane (lab analyses, 1-$\sigma$ of 7 ppb) and $\delta^{18}O_{atm}$ (1-$\sigma$ of 0.01 ‰)





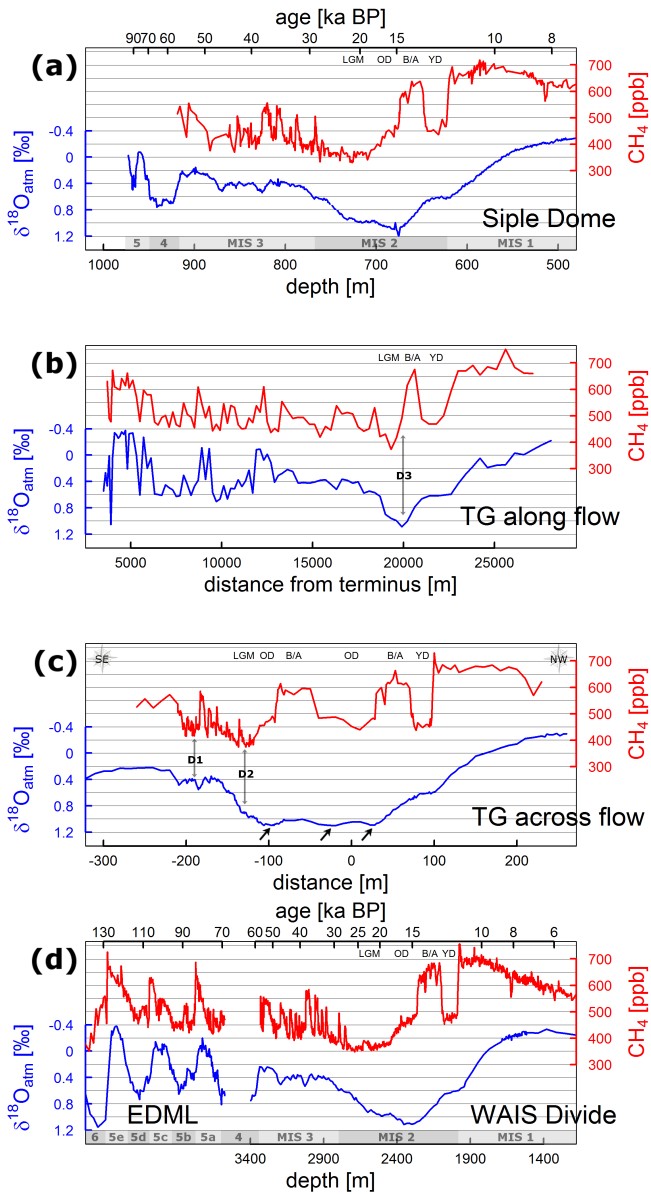

**Figure 3.** $\delta^{18}O_{atm}$ (blue, note reversed scale) and methane (red) records from deep ice cores (Siple Dome (Brook et al., 2005; Severinghaus et al., 2009), WD (Buizert et al., 2015), EDML (Capron et al., 2010; Schilt et al., 2010)) and from Taylor Glacier (across-flow, along-flow). TG along-flow methane data is based on lab analyses only, whereas the across-flow data is a combination of lab and field analyses. For TG across-flow, the 0 m point is arbitrarily chosen at a conspicuous circular feature (Fig. 5a), and the threefold $\delta^{18}O_{atm}$ maximum is highlighted with arrows. The approximate locations of the three deep cores D1, D2, and D3 are shown in (b) and (c). The intersection point of the two TG sampling lines is at -240 m on the across-flow scale, and at 13.5 km on the along-flow scale. Distinct climate periods of the deglaciation (top of each panel) and Marine Isotope Stages (MIS, bottom) are indicated.





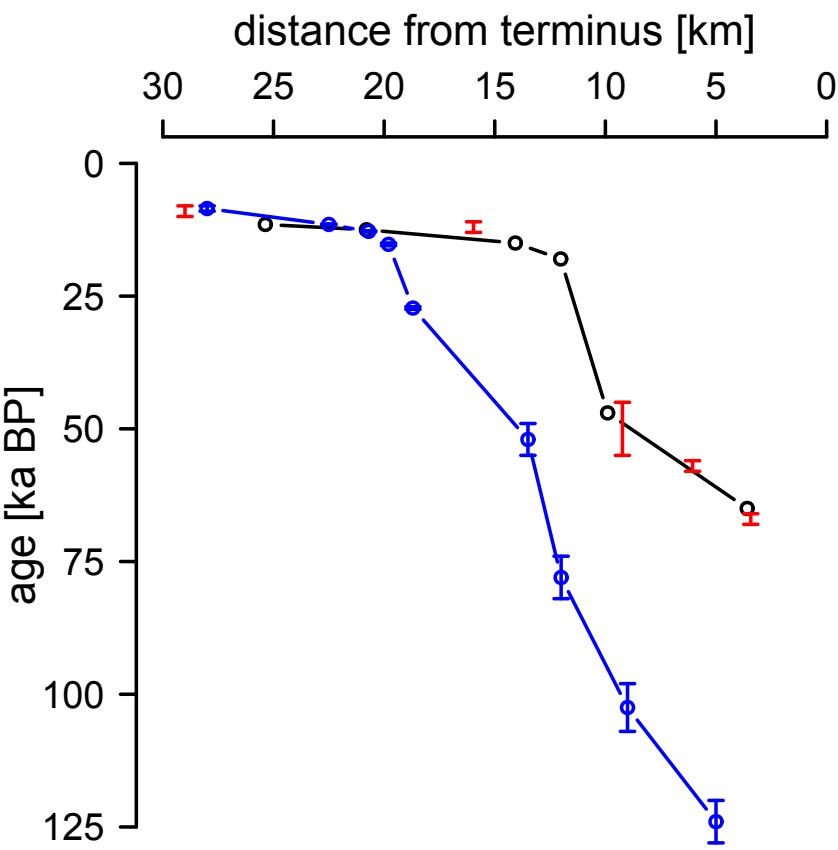

**Figure 4.** Distance age estimates for the Taylor Glacier ablation zone. (Black) Aciego et al. (2007)'s age model based on matching $\delta$D to Taylor Dome and supported by five $\delta^{18}O_{atm}$ age estimates (red, from their Fig. 5). 1 km was added to Aciego et al. (2007)'s distance scale to account for the fact that their sampling line ended approximately 1 km from the glacier terminus. (Blue) Our along-flow profile age model, based on $\delta^{18}O_{atm}$ and methane concentration. Note that the sampling locations were not identical in both studies and therefore the results are not directly comparable.




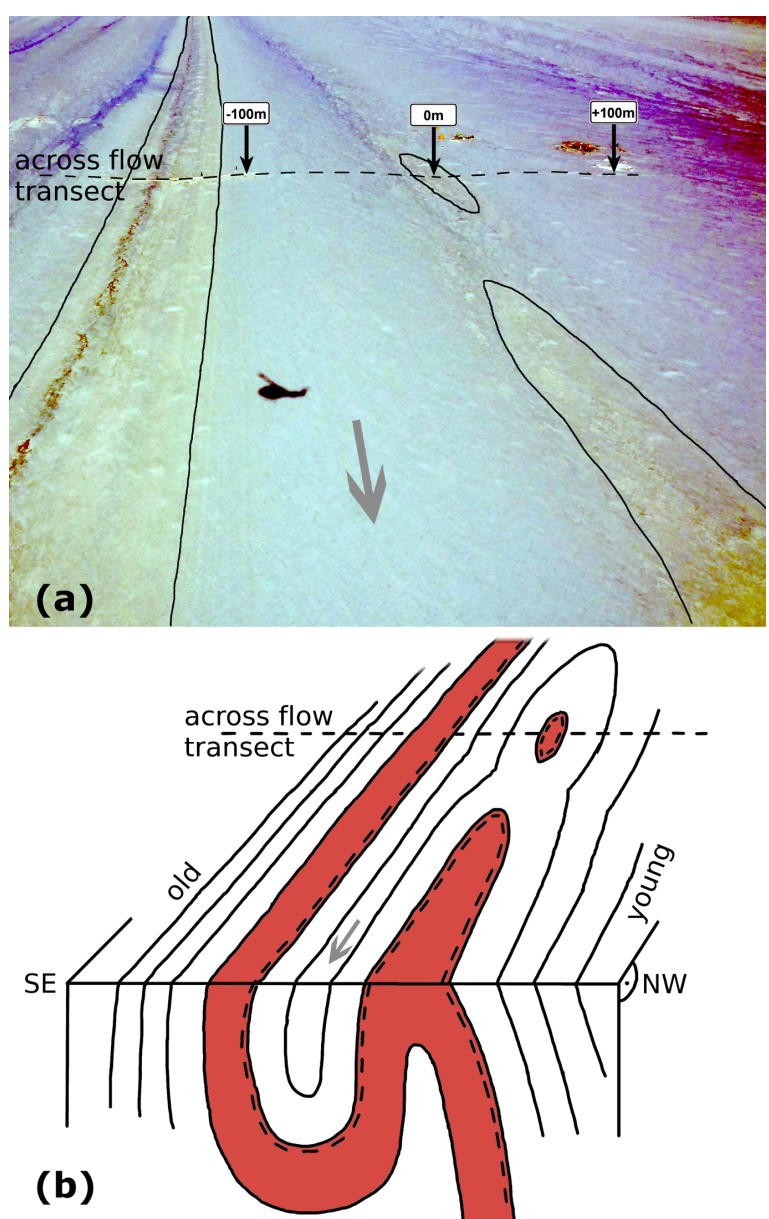

**Figure 5.** Folding on Taylor Glacier. **(a)** Photo of the across-flow transect sampling area showing two limbs of the dusty band. Color settings were adjusted to enhance the color differences. The solid black line marks the approximate boundary of the dusty band. The dashed black line marks the across-glacier transect including distance markers. The camp is visible in the upper right. Glacier flow is from top to bottom and indicated by gray arrows. For scale, the south-eastern limb of the dusty band, on the left in the photo, is ∼60 m wide. Photo by Ed Brook. **(b)** Sketch of the z-fold, showing its surface expression and a cross section. The dusty band is shown in brown.





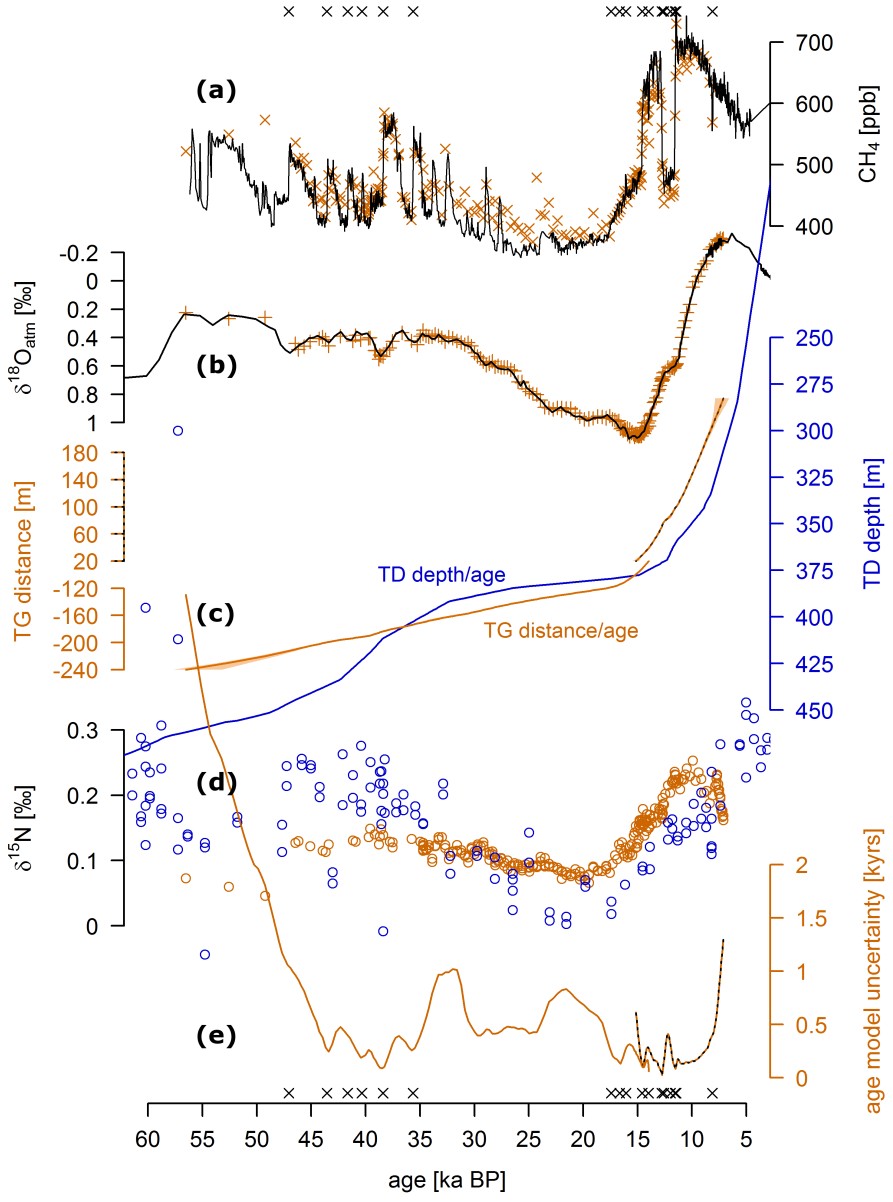

**Figure 6. (a)** Methane records from WAIS Divide (black, Buizert et al. (2015)) and Taylor Glacier (orange, this study and Bauska et al. (2016)). Taylor Glacier methane data are a combination of field and lab analyses. WAIS Divide data are on the WD2014 time scale. **(b)** $\delta^{18}O_{atm}$ records from WAIS Divide (black) and Taylor Glacier (orange). **(c)** Taylor Dome depth/age curve (blue, from Steig et al. (1998) and Sucher (1997)), and Taylor Glacier distance/age relation (orange). The Taylor Dome time scale is based on methane synchronization to GISP2 (Brook et al., 2000; Ahn and Brook, 2007). **(d)** $\delta^{15}N$ from Taylor Dome (Sucher, 1997) and Taylor Glacier (orange). **(e)** Width of the 90% confidence interval of the TG gas model age distribution. Black crosses on the top and bottom of the plot show fixed tie points used in the WAIS Divide–Taylor Glacier synchronization (cf. Table 1).

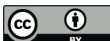



**Table 1.** Fixed tie points used in the gas age model synchronizing Taylor Glacier to WAIS Divide. Note that two tie points (at 2207.3 m and 2258.9 m WAIS Divide depth) are overlapping, i.e. are used both for the "old" and the "young" side of the transect. WAIS Divide is on the WD2014 age scale (Buizert et al., 2015).

| Gas age [yrs BP] | WAIS Divide depth [m] | TG across-flow distance [m] | TG across-flow range [m] | Type of tie point |
|---|---|---|---|---|
| 8,123 | 1,620.0 | 220 | 210 to 230 | $CH_4$ minimum |
| 11,451 | 1,972.0 | 100 | 99 to 101 | $CH_4$ max and $CO_2$ inflection |
| 11,548 | 1,983.4 | 98 | 97 to 99 | $CH_4$ transition midpoint (YD–PB) |
| 11,965 | 2,023.5 | 86 | 83 to 90 | $CO_2$ transition midpoint |
| 12,583 | 2,079.5 | 78 | 76 to 80 | $CH_4$ minimum |
| 12,765 | 2,096.2 | 71.5 | 71 to 72 | $CH_4$ transition midpoint (BA–YD) |
| 13,989 | 2,207.3 | 40 | 36 to 42 | $CH_4$ minimum |
| 14,571 | 2,258.9 | 29 | 28 to 30 | $CH_4$ transition midpoint (OD–BA) |
| 13,989 | 2,207.3 | -82 | -84 to -80 | $CH_4$ minimum |
| 14,571 | 2,258.9 | -91 | -92 to -90 | $CH_4$ transition midpoint (OD–BA) |
| 16,062 | 2,370.0 | -110 | -111 to -108 | $CO_2$ and $CH_4$ peak |
| 16,640 | 2,399.5 | -114.5 | -115 to -114 | $CO_2$ transition midpoint |
| 17,418 | 2,432.0 | -118.5 | -119 to -118 | $CO_2$ inflection point |
| 35,642 | 2,958.8 | -173.5 | -174 to -173 | $CH_4$ transition midpoint (DO 7) |
| 38,386 | 3,021.5 | -183.5 | -183.7 to -183.3 | $CH_4$ transition midpoint (DO 8) |
| 40,336 | 3,066.6 | -191.75 | -192.0 to -191.5 | $CH_4$ transition midpoint (DO 9) |
| 41,650 | 3,094.3 | -194.5 | -195 to -194 | $CH_4$ transition midpoint (DO 10) |
| 43,547 | 3,130.5 | -199.5 | -200 to -199 | $CH_4$ transition midpoint (DO 11) |
| 47,061 | 3,195.2 | -211 | -215 to -210 | $CH_4$ transition midpoint (DO 12) |





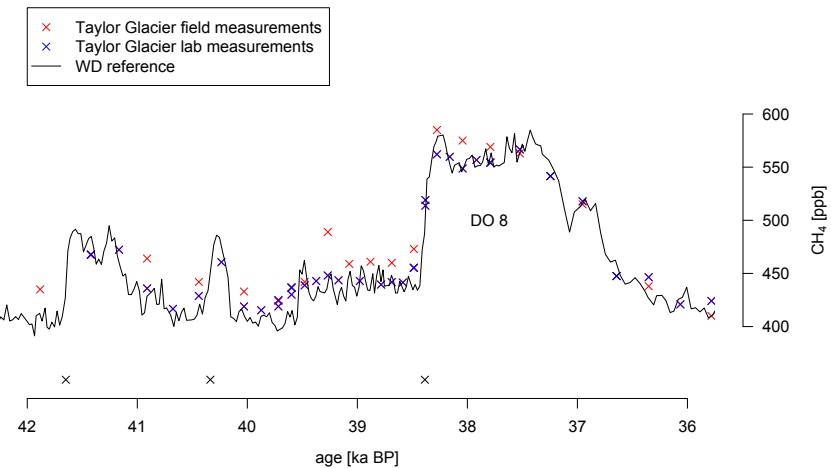

**Figure A1.** Methane data from Taylor Glacier field analyses (red crosses), lab analyses (blue crosses), and the WD reference record (black line) for a well-sampled time period during MIS 3 including Dansgaard/Oeschger (DO) event 8. Black crosses on the bottom of the plot show fixed tie points used in the WAIS Divide–Taylor Glacier synchronization (Table 1).

