# Peer review of "Atmospheric gas records from Taylor Glacier, Antarctica, reveal ancient ice with ages spanning the entire last glacial cycle"

_Climate of the Past, 2017_

## Referee Comment (RC1) · T. Sowers (Referee) · 28 Mar 2017

This paper is very well written and should be published with minor revisions. In general, the length of the paper was a bit excessive but warranted I think in this case, as it presents the first overall survey of the Taylor Glacier mining expeditions and results.

Couple of thoughts came to mind in reviewing the discussion of Figure 6. First, the $\delta$18Oatm data is superb in the agreement with WAIS. This makes for a very convincing stratigraphy. However, the CH4 data is less impressive. For the bulk of the glacial period, TG CH4 data are elevated above WAIS. But, during the transition, TG CH4 data are lower than WAIS for the most part (Bolling/Alerod and PreBoreal for sure). What is the significance of this apparent mismatch and how important is this in constructing

the stratigraphy?

The end of the "easily interpretable stratigraphy" seems to be 47ka. There you have a single point (around 13km) where the CH4 value is high, the d18Oatm is Ok but the d15N value is very low ($\sim$0.07). I'd argue that this particular sample is the beginning of the unconstrained region where more work is needed to verify the integrity of the record down to the terminus. I think it would be fair to place a vertical dashed line at 47ka denoting the switch from well constrained stratigraphy to the dark side.

---

## Referee Comment (RC2) · Anonymous Referee #2 · 12 Apr 2017

This manuscript deals with the challenges of dating a horizontal ice core. Taylor glacier served as a climate archive for research that demands large samples. Proper dating is a prerequisite for a climatic interpretation of the results. Dating is achieved via "global" atmospheric gases. The glaciological interpretation involves a lot of hand waving but this is in the nature of it. The manuscript is well written and I have only very few comments that can easily be implemented in minor revisions. 1) The synchronization is done with atmospheric gases. Therefore, it is on the gas time scale. The ice of the sample is older than the gas. This fact needs to be mentioned in the introduction. I would also appreciate a bulk number on the order magnitude of Dage. 2) Other than d18Oatm, methane has a concentration gradient over the globe. "Value-matching"

[Figure]

(p2, line 32) of methane is only possible since Taylor glacier, WAIS, and EDC are in latitudinal proximity. This needs to be mentioned. 3) The manuscript is well referenced but I miss Chappellaz et al., J Geophys Res 102, 26547-26558 (1997) who was the first to my knowledge that has used CH4 and d18Oatm to reconstruct the chronology of an ice core.

———————————————————

---

## Author Response (AR1)

Reply to comments provided by **T. Sowers** on 'Atmospheric gas records from Taylor Glacier, Antarctica, reveal ancient ice with ages spanning the entire last glacial cycle' by D. Baggenstos et al.

We thank Todd Sowers for his positive assessment of our study.
The main point raised concerns the quality of the methane data. First off, any offsets between Taylor Glacier and WAIS do not affect the construction of the time scale/stratigraphy, because in our age model we are only using the fast methane transitions at the beginning of D-O events and during the termination, which can be easily identified also in less than perfect quality methane data. However, the methane record does raise the question whether there are systematic offsets between Taylor Glacier ice and WAIS. We think that most of the apparent mismatch can be attributed to the fact that some of the data (and most of it in the glacial) was produced by a field deployable methane extraction system that is not calibrated as well as lab based systems and suffers from generally less than ideal conditions to do precise measurements in the field. As shown in the appendix, glacial ice measured in the lab does not exhibit the apparent offset. We would point this out more clearly in a revised manuscript.

The second comment regarding the end of the easily interpretable stratigraphy is fair and we will add a line denoting this to the figure as suggested.

Reply to comments provided by **anonymous reviewer** on 'Atmospheric gas records from Taylor Glacier, Antarctica, reveal ancient ice with ages spanning the entire last glacial cycle' by D. Baggenstos et al.

We thank the reviewer for his positive assessment of our study.
The comments provided will be addressed in a revised manuscript and a reference to the Chappellaz paper will be added. Regarding the second point raised by the reviewer (value-matching), we would like to point out that value-matching is only performed with the oxygen isotope data, which has no appreciable inter-hemispheric gradient. For methane only distinct rapid transitions are used in the age model. We will clarify this in the revisions.

[revised manuscript text omitted]

15  transitions as tie points to anchor the synchronization algorithm, and do not perform any value matching with methane.

*Competing interests.*  The authors declare that they have no conflict of interest.

*Acknowledgements.*  We thank Kurt Cuffey for helpful discussions on glaciology and the logistics of living on Taylor Glacier. The US Antarctic Program provided outstanding logistical support. IDDO (Ice Drilling Design and Operations) provided the drilling systems. The Polar Geospatial Center (PGC) provided satellite imagery. We thank Kate Koons, Chandra Llewellyn, Paul Rose, Bija Sass and Martha

20  Story for keeping everybody safe, fed, and happy. Michael Dyonisius, Xavier Fain, Ben Hmiel, Tanner Kuhl, Robb Kulin, Logan Mitchell, Avery Palardy and Adrian Schilt all helped with sampling in the field. We thank Ross Beaudette for laboratory assistance. Todd Sowers and an anonymous referee reviewed the manuscript and provided helpful feedback. This work is supported through NSF Polar Awards 0839031, 1246148, 1245821 and 1245659. Further support came from the Marsden Fund Council from New Zealand Government funding, administered by the Royal Society of New Zealand and NIWA under Climate and Atmosphere Research Programme CAAC1504 (HS).

**References**

[revised manuscript text omitted]
 1). The dashed gray line marks the switch from well-constrained stratigraphy to older ice where more work is needed to construct a robust age scale. 24

**Table 1.** Fixed tie points used in the gas age model synchronizing Taylor Glacier to WAIS Divide. Note that two tie points (at 2207.3 m and 2258.9 m WAIS Divide depth) are overlapping, i.e. are used both for the "old" and the "young" side of the transect. WAIS Divide is on the WD2014 age scale (Buizert et al., 2015).

| Gas age [yrs BP] | WAIS Divide depth [m] | TG across-flow distance [m] | TG across-flow range [m] | Type of tie point |
|---|---|---|---|---|
| 8,123 | 1,620.0 | 220 | 210 to 230 | $CH_4$ minimum |
| 11,451 | 1,972.0 | 100 | 99 to 101 | $CH_4$ max and $CO_2$ inflection |
| 11,548 | 1,983.4 | 98 | 97 to 99 | $CH_4$ transition midpoint (YD–PB) |
| 11,965 | 2,023.5 | 86 | 83 to 90 | $CO_2$ transition midpoint |
| 12,583 | 2,079.5 | 78 | 76 to 80 | $CH_4$ minimum |
| 12,765 | 2,096.2 | 71.5 | 71 to 72 | $CH_4$ transition midpoint (BA–YD) |
| 13,989 | 2,207.3 | 40 | 36 to 42 | $CH_4$ minimum |
| 14,571 | 2,258.9 | 29 | 28 to 30 | $CH_4$ transition midpoint (OD–BA) |
| 13,989 | 2,207.3 | -82 | -84 to -80 | $CH_4$ minimum |
| 14,571 | 2,258.9 | -91 | -92 to -90 | $CH_4$ transition midpoint (OD–BA) |
| 16,062 | 2,370.0 | -110 | -111 to -108 | $CO_2$ and $CH_4$ peak |
| 16,640 | 2,399.5 | -114.5 | -115 to -114 | $CO_2$ transition midpoint |
| 17,418 | 2,432.0 | -118.5 | -119 to -118 | $CO_2$ inflection point |
| 35,642 | 2,958.8 | -173.5 | -174 to -173 | $CH_4$ transition midpoint (DO 7) |
| 38,386 | 3,021.5 | -183.5 | -183.7 to -183.3 | $CH_4$ transition midpoint (DO 8) |
| 40,336 | 3,066.6 | -191.75 | -192.0 to -191.5 | $CH_4$ transition midpoint (DO 9) |
| 41,650 | 3,094.3 | -194.5 | -195 to -194 | $CH_4$ transition midpoint (DO 10) |
| 43,547 | 3,130.5 | -199.5 | -200 to -199 | $CH_4$ transition midpoint (DO 11) |
| 47,061 | 3,195.2 | -211 | -215 to -210 | $CH_4$ transition midpoint (DO 12) |

[Figure]

**Figure A1.** Methane data from Taylor Glacier field analyses (red crosses), lab analyses (blue crosses), and the WD reference record (black line) for a well-sampled time period during MIS 3 including Dansgaard/Oeschger (DO) event 8. Black crosses on the bottom of the plot show fixed tie points used in the WAIS Divide–Taylor Glacier synchronization (Table 1).